# Population homogeneity for the antibody response to COVID-19 BNT162b2/Comirnaty vaccine is only reached after the second dose across all adult age ranges

João Faro-Viana [1,6], Marie-Louise Bergman [2,6], Lígia A. Gonçalves[2], Nádia Duarte [2], Teresa P. Coutinho [3], Patrícia C. Borges[2], Christian Diwo[2], Rute Castro [4], Paula Matoso[2], Vanessa Malheiro[2], Ana Brennand[2], Lindsay Kosack [2], Onome Akpogheneta[2], João M. Figueira[1], Conceição Cardoso[1], Ana M. Casaca[1], Paula M. Alves [4,5], Telmo Nunes [3], Carlos Penha-Gonçalves [2✉] & Jocelyne Demengeot [2✉]

While mRNA vaccines are administrated worldwide in an effort to contain the COVID-19 pandemic, the heterogeneity of the humoral immune response they induce at the population scale remains unclear. Here, in a prospective, longitudinal, cohort-study, including 1245 hospital care workers and 146 nursing home residents scheduled for BNT162b2 vaccination, together covering adult ages from 19 to 99 years, we analyse seroconversion to SARS-CoV-2 spike protein and amount of spike-specific IgG, IgM and IgA before vaccination, and 3-5 weeks after each dose. We show that immunogenicity after a single vaccine dose is biased to IgG, heterogeneous and reduced with increasing age. The second vaccine dose normalizes IgG seroconversion in all age strata. These findings indicate two dose mRNA vaccines is required to reach population scale humoral immunity. The results advocate for the interval between the two doses not to be extended, and for serological monitoring of elderly and immunosuppressed vaccinees.

[1] CHLO, Centro Hospitalar de Lisboa Ocidental, Serviço de Patologia Clínica, Lisbon 1449-005, Portugal. [2] IGC, Instituto Gulbenkian de Ciência, Oeiras 2780-156, Portugal. [3] CIISA, Centre for Interdisciplinary Research in Animal Health, Faculty of Veterinary Medicine, University of Lisbon, Lisbon 1300-477, Portugal. [4] IBET, Instituto de Biologia Experimental e Tecnológica, Oeiras 2780-901, Portugal. [5] ITQB NOVA, Instituto de Tecnológia Química e Biológica António Xavier, Universidade Nova de Lisboa, Oeiras 2780-157, Portugal. [6] These authors contributed equally: João F. Viana, Marie-Louise Bergman. ✉email: cpenha@igc.gulbenkian.pt; jocelyne@igc.gulbenkian.pt

Authorization for emergency use of two mRNA COVID-19 vaccines, both encoding the most immunogenic protein of SARS-CoV-2, spike, was conceded in late 2020 by regulatory agencies such as FDA and EMA. These authorizations were based on results of phase 3 clinical trials that demonstrated high standards of safety and high levels of efficacy in preventing symptomatic SARS-CoV-2 infections[1,2]. While these vaccines are introduced around the world and administered to millions of people, there is a growing and acute need to evaluate their effectiveness at the population level, an endeavor that may require months of epidemiological studies. Insufficient attention has been given to whether immune responses triggered by mRNA vaccines encoding SARS-CoV-2-spike are homogenously robust. Age and gender are expected factors of variability. Immune responses deteriorate with age, underlying the increased burden of infectious disease, including COVID-19[3], in older people as well as impaired responses to vaccine challenge[4]. Sex differences have been described in immunity to multiple vaccines in both children and adults, and antibody responses to vaccines are frequently higher in females than males[5].

To date, humoral immune responses have been seldom measured upon mRNA COVID-19 administration, and when this was the case, limited to the IgG class and concerned rather small groups of participants, ranging from $n = 8$–51[2,6–15]. Immunogenicity of mRNA COVID-19 vaccines and their inter-individual variation can be easily monitored in medium to large cohorts by measuring serum reactivities to the vaccine antigen or part of it. Notably, the receptor-binding domain (RBD) of spike contains the amino-acids motifs permitting SARS-CoV-2 binding to the angiotensin-converting enzyme 2 (ACE2) receptor, a prerequisite for infection, and serum reactivity to this region encompasses neutralizing activity[16]. Anti-spike immunoglobulins are also expected to mediate viral particle removal through antibody-mediated opsonization and phagocytosis, and through the recruitment of the complement system. Beyond their direct functionality, vaccine-specific antibodies are markers of adaptive immunity responses[17,18].

The COVID-19 vaccination campaign in Portugal was initiated in late December 2020 coinciding with a peak of disease transmission which reached 131 new daily cases per 100,000 inhabitants and caused an unprecedented demand for hospital care. The vaccination roll-out started with hospital healthcare professionals at the COVID-19 response frontline, soon followed by residents in nursing homes.

In this work, we report on the humoral response to BNT162b2 mRNA COVID-19 (Comirnaty, Pfizer/BioNTech) vaccination in healthcare professionals and in nursing home residents. We show immunogenicity after a single vaccine dose is biased to IgG, heterogeneous and reduced with increasing age. The second vaccine dose normalizes IgG seroconversion in all age strata. These findings indicate two-dose mRNA vaccines are required to reach population-scale humoral immunity.

## Results

**Enrollment**. The study followed 1245 healthcare workers (HCW cohort) and 146 nursing home residents (NHR cohort) vaccinated with BNT162b2 mRNA (Comirnaty, Pfizer/BioNTech) (Fig. 1). Prior COVID-19 diagnosis was an exclusion criterion, in accordance with the national vaccination plan, and reported cases were restricted to the HCW cohort. Venous blood was collected on the day of 1st vaccine dose administration (time 0, t0), 3–5 weeks later at the day of the 2nd injection (t1), and 3 weeks after the 2nd dose administration (t2). Both cohorts present a biased sex ratio (females 79% in HCW and 74% in NHR), as is common in these populations in occident, and together encompass a broad age

range (median [age range]: HCW 43 [19–70] and NHR 87 [70–99]). To identify participants with unknown prior infection, the entire HCW cohort was tested at t0 for serum reactivity against SARS-CoV-2 nucleocapsid (N), detecting 23 positive cases (2%). In the NHR cohort, epidemiologic surveillance by the health system evidenced no COVID-19 cases prior to vaccination, and all samples collected at day 0 tested negative for anti-spike reactivities. Collected samples were analysed for bulk reactivity against SARS-CoV-2-RBD using a commercial ECLIA (HCW only), and for isotype-specific (IgG, IgM, and IgA) anti-SARS-CoV-2-spike using an in-house ELISA assay.

**Infection**. During the course of the study, 43 participants were diagnosed with COVID-19 by RT-PCR on the nasopharyngeal swab (Fig. 1). Of these, 38 were infected in the interval between the two vaccine doses (HCW 31/1245, 2.5% and NHR 7/146, 4.8%, median [IQR] 1.7 [1.14–2.14] weeks post-1st dose for the 2 cohorts). An additional 14 HCW showed de novo SARS-CoV-2 N antigen reactivity at t1 and/or t2. Diagnosed and inferred cases of COVID-19 post t0 were excluded from the following immunogenicity analysis.

**Immunogenicity**. To directly determine the immunogenicity of the BNT162b2 vaccine, SARS-CoV-2 naïve participants, defined as negative for serum reactivity anti-SARS-CoV-2 N (HCW, $n = 948$) or spike (NHR, $n = 118$), were first analysed using a binary classification (Fig. 2a). As expected, seroconversion was the rule in HCW, with bulk anti-RBD reactivity detected at similar frequency whether at 3 weeks post 1st or 2nd injection (99%; 95% CI 98–99 at t1, 100%; 99–100 at t2). Isotype class analysis of anti-spike antibodies revealed a heterogeneous response at t1, with 89% (95% CI 87–91) positivity for IgG, 41% (38–44) for IgM, and 69% (66–72) for IgA. Increased positivity at t2 was limited to the IgG class, reaching 100% (99–100). In contrast, seroconversion at t1 was poor in the NHR cohort with only 25% (18–34) positivity for IgG, 3% (1–7) for IgM, and 36% (28–45) for IgA, while the 2nd vaccine dose resulted in 95% (89–98) positivity for IgG reactivities.

Quantitative analysis (Fig. 2b and supplementary Table 1) revealed very large inter-individual heterogeneity in the amplitude of anti-RBD Ig and anti-spike IgG responses at t1, covering the entire dynamic range of each assay (median [IQR]: HCW 1.44 [1.26–1.56]; NHR 0.50 [0.35–1.06], for anti-S IgG at t1). Large heterogeneity was also observed for anti-spike IgM and IgA levels. Strikingly, the 2nd vaccine dose resulted in major increment and homogenization to high values of anti-RBD Ig and anti-spike IgG responses, with measurements reaching saturation for the vast majority of participants in HCW as in NHR (median [IQR]: HCW 1.83 [1.72–1.92] and NHR 1.83 [1.68–1.96], for anti-spike IgG at t2). The median anti-spike IgG response was estimated to correspond to titres of at least $1 \times 10^4$ at t2 versus $1 \times 10^3$ and $1 \times 10^2$ at t1 in HCW and NHR, respectively, indicating the 2nd dose provides an increment higher than 10-fold. In contrast, the 2nd vaccine dose does not improve anti-spike-specific IgM and IgA responses, or only mildly as for few NHR participants.

**Non-responders**. In both cohorts we identified non-responders, defined as not reaching anti-spike IgG level of positivity after 2 vaccine doses (1/948 naïve HCW, 0.1%; 6/118 NHR, 5%). The HCW non-responder, also classified as negative for anti-RBD, was treated for Rheumatoid Arthritis (leflunomide, steroids, and methotrexate), and did not respond to a previous Hepatitis B vaccination. Lack of responsiveness in 6 NHR participants, 3 males and 3 females with 84–91 years (median 89), showed no apparent association with frailty or medication.

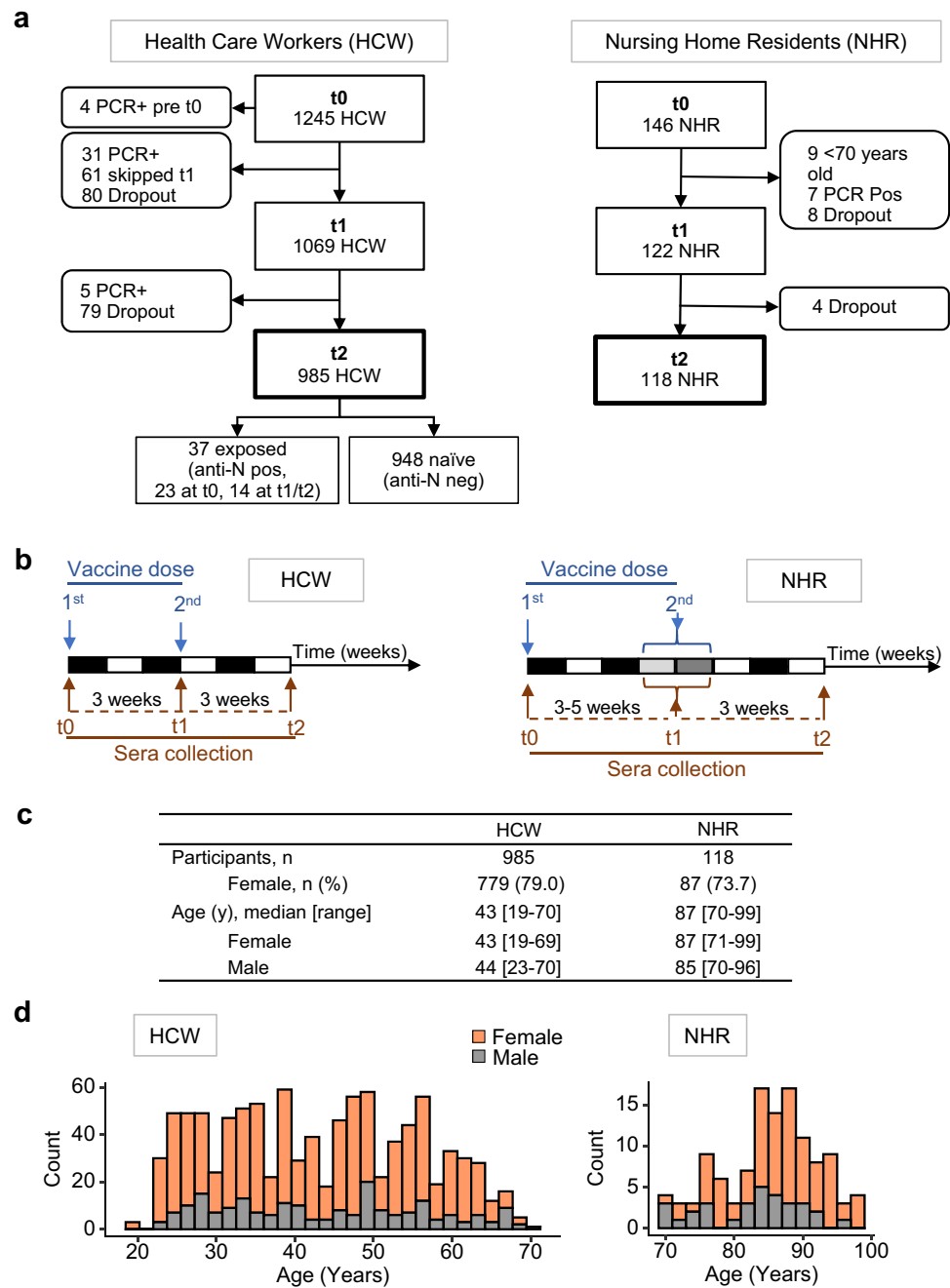

**Fig. 1 Cohort specification.** Hospital healthcare workers (HCW) and nursing home residents (NHR) donated blood samples before vaccination with BNT162b2 RNA (t0), 3–5 weeks after the 1st dose (t1) and 3 weeks after the 2nd dose (t2). **a** Enrollment and funnelling of participants, showing concordance to the study design (straight boxes), and exclusion criteria to the main immunogenicity analysis or dropouts (rounded boxes). **b** Collection Schedule. **c** Age and sex distribution. **d** Stratification by a 2-year interval. Source data are provided as a Source Data file.

**Cumulative age effect**. The parallel analysis of the HCW and NHR cohorts revealed a dramatic effect of advanced age on the magnitude of the antibody response at t1, for all three anti-spike isotypes (Fig. 2). Stratified analysis by age groups (in 10-year bins) of the HCW cohort revealed an increasingly negative effect of age at t1, evident for bulk anti-RBD reactivity and for anti-spike IgG and IgM, but not IgA levels (Fig. 3 and supplementary Table 2). While age stratification within the NHR cohort ([70–85] and [86–99] years) was not informative, the youngest age strata of the NHR cohort scored lower than the oldest age strata of the HCW cohort, for anti-spike IgG, IgM and IgA (median [IQR] [age strata] for anti-spike IgG: HCW 1.30 [1.00–1.53] [60–69]; NHR 0.53 [0.42–1.21] [70–85]; for anti-spike IgM: HCW 0.72 [0.54–0.98] [60–69]; NHR 0.36 [0.26–0.48] [70–85] and for anti-spike IgA: HCW (1.11 [0.88–1.20] [60–69]; NHR 0.82 [0.59–1.12]).

**Sex effect**. Stratification of the cohorts by sex evidenced marginal effect (Fig. 4 and supplementary Tables 1 and 2). After 1st dose administration, males presented lower anti-RBD and anti-spike responses, only in the age stratum 60–69 years, a result that affected also the frequency of IgG seroconversion in this age range (positivity at t1, frequency (95% CI) 82% (71–89) for females, 55.2% (36–73) for males). In the older NHR cohort, IgG levels were marginally higher in females than in males, and only

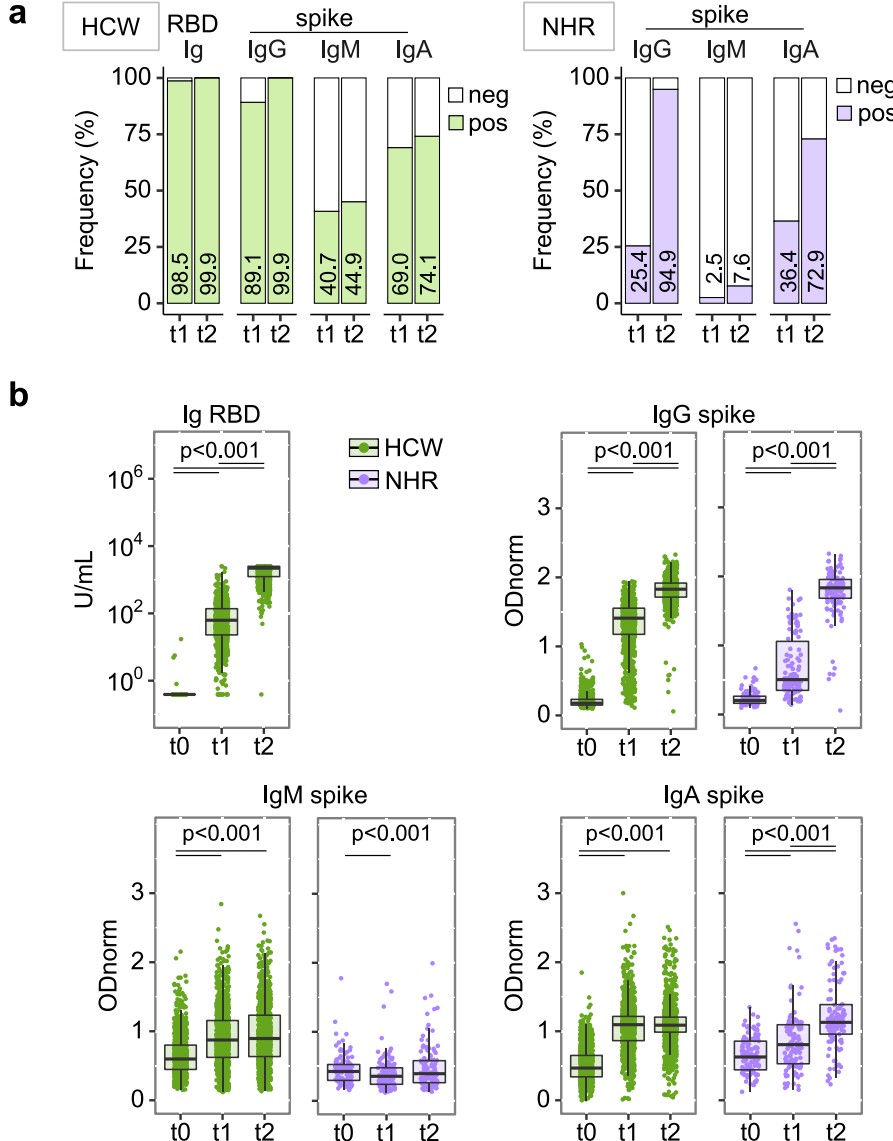

**Fig. 2 Heterogenous anti-SARS-CoV-2-spike reactivities induced by vaccination.** Sera collected as in Fig. 1c were analysed for anti-RBD Ig (ECLIA) and anti-full-length spike protein IgG, IgM, and IgA (ELISA). Individuals positive for reactivities against SARS-CoV-2 N antigen were removed from the dataset. Health Care Workers (HCW) $n = 948$, in green, and Nursing Home Residents (NHR) $n = 118$, in purple. **a** Seroconversion defined by frequency of samples testing positive (colored bars) at the indicated day and assays. Respective values are indicated inside each bar. **b** Semi-quantitative measurements. Data points represent individual participants, boxes denote interquartile range, horizontal line represent the median, and whiskers denote the minimum and maximum values below or above the median at 1.5 times the interquartile range. Note the y scale differs for the anti-RBD ECLIA and the anti-spike ELISA data. Index $\geq 0.8$ and ODnorm $\geq 1$ define positivity in (**a**). Quade test for group difference across time points, $p < 2.2 \times 10^{-16}$ for all panels except NHR IgM where $p = 0.001096$. Pairwise Wilcoxon signed-rank test over time, two-sided, with $p$-value adjustment (Benjamini–Hochberg method). HCW: for anti-RBD Ig and anti-spike $p < 2.2 \times 10^{-16}$ (t0/t1, t0/t2, t1/t2); for anti-spike IgM $p < 2.2 \times 10^{-16}$ (t0/t1, t0/t2) and $p = 0.4$ (t1/t2); for anti-spike IgA $p < 2.2 \times 10^{-16}$ (t0/t1, t0/t2) and $p = 0.39$ (t1/t2). NHR: for anti-spike IgG $p = 1.7 \times 10^{-06}$ (t0/t1, t1/t2) and $p = 6.4 \times 10^{-05}$ (t0/t2); for anti-spike IgM $p = 0.038$ (t0/t1), $p = 1.000$ (t0/t2) and $p = 0.318$ (t1/t2); for anti-spike IgA $p = 0.0014$ (t0/t1, t0/t2) and $p = 0.0039$ (t1/t2). Significant $p$-values are indicated in each panel. Wilcoxon rank-sum test for difference between HCW and NHR at t1, $p = 4.40 \times 10^{-16}$ for IgG, $p < 2.2 \times 10^{-16}$ for IgM and $p = 1.7056 \times 10^{-13}$ for IgA; at t2 $p = 0.9796$ for IgG, $p < 2.2 \times 10^{-16}$ for IgM and $p = 0.01084$ for IgA. Source data are provided as a Source Data file.

at t2. Despite the overall higher immuno-competence of the youngest age stratum, levels of specific reactivities were still strikingly spread at t1 in this age group (e.g. titre ranges were estimated from $1 \times 10^2$ to $1 \times 10^4$ or higher, for anti-spike IgG in the [19–29] stratum). We excluded concerns of RNA vaccine stability, as stratification by calendar days of the 1st vaccine dose administration showed no effect on antibody levels.

**Extended prime-boost interval.** The NHR participants received the 2nd vaccine dose either 3 ($n = 44$) or 5 ($n = 74$) weeks after

administration of the 1st dose. In both subgroups, the 2nd dose boosted the anti-spike IgG levels to high values, with an apparent higher amplitude after the shorter time interval, and resulted in a similar frequency of positivity (94.1% for 3- and 93.2% for 5-week interval) (Supplementary Fig. S1). As a possible confounding factor, sex distribution was different between the two groups, with more males in the 5-weeks group (82% females in the 3- and 69% in the 5-weeks group), while the average age was similar. This data supports the notion that a shorter prime-boost interval does not prevent a robust recall response.

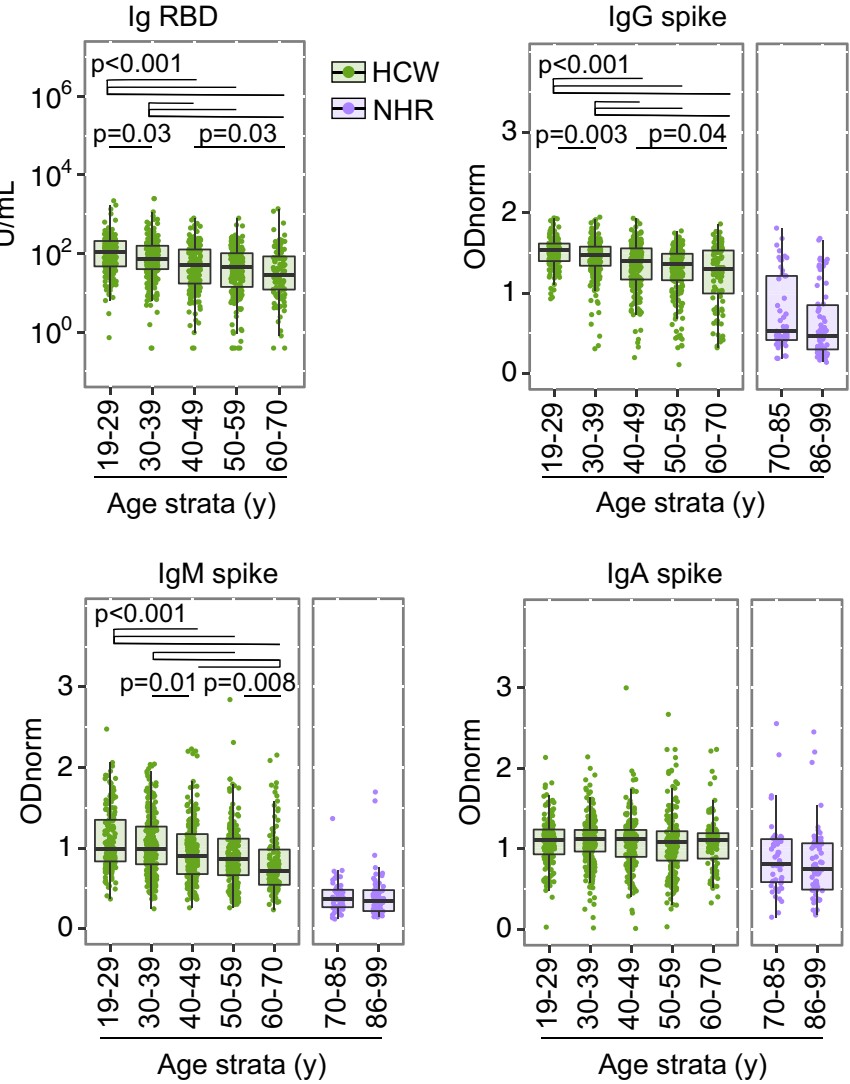

**Fig. 3 Cumulative effect of age on anti-SARS-CoV-2-spike reactivities induced by vaccination.** Shown are semi-quantitative measurements in the Health Care Workers (HCW) $n = 948$, in green, and Nursing Home Residents (NHR) $n = 118$, in purple, cohorts at t1, as in Fig. 2B, now stratified by age group in 10 or 15 year intervals ([19–29]y $n = 171$; [30–39]y $n = 251$; [40–49]y $n = 204$; [50–59]y $n = 216$; [60–70]y $n = 107$; [70–85]y $n = 51$; [86–99]y $n = 66$). Data points represent individual participants, boxes denote interquartile range, horizontal lines represent the median, and whiskers denote the minimum and maximum values below or above the median at 1.5 times the interquartile range. Kruskal–Wallis test for group difference across HCW age groups, $p = 1.875 \times 10^{-15}$ for anti-RBD Ig, $p < 2.2 \times 10^{-16}$ for anti-spike IgG, $p = 4.385 \times 10^{-13}$ for anti-spike IgM, $p = 0.55$ for anti-spike IgA. Significant $p$-values from Pairwise Wilcoxon rank-sum comparison, two-sided, with $p$-value adjustment (Benjamini–Hochberg method), are indicated in each HCW panel. Wilcoxon rank-sum test comparing antibody levels in HCW age group [60–70]y and NHR age group [70–85]y at t1, $p = 3.405 \times 10^{-08}$ for IgG, $p < 2.2 \times 10^{-16}$ for IgM and $p = 0.0002751$ for IgA. in HCW $p$-values for comparisons [30–39] vs [19–29]; [40–49] vs [19–29] and vs [30–39]; [50–59] vs [19–29], vs [30–39] and vs [40–49]; [60–70] vs [19–29], vs [30–39], vs [40–49], and vs [50–59] are respectively: for anti-RBD Ig, $p = 0.03345$; $p = 5.6 \times 10^{-07}$ and $p = 0.00065$; $p = 5.7 \times 10^{-10}$, $p = 5.1 \times 10^{-07}$, and $p = 0.16287$; $p = 1.6 \times 10^{-09}$, $p = 3.8 \times 10^{-07}$, $p = 0.02502$, and $p = 0.16287$. For anti-spike IgG, $p = 0.0031$; $p = 5.6 \times 10^{-07}$ and $p = 0.00065$; $p = 2.5 \times 10^{-14}$, $p = 9.3 \times 10^{-09}$ and $p = 0.0732$; $p = 2.4 \times 10^{-09}$, $p = 4.1 \times 10^{-06}$, $p = 0.0370$ and $p = 0.4323$. For anti-spike IgM, $p = 0.26467$; $p = 0.00068$ and $p = 0.00815$; $p = 6.8 \times 10^{-07}$, $p = 2.2 \times 10^{-05}$, and $p = 0.13583$; $p = 6.6 \times 10^{-10}$, $p = 1.4 \times 10^{-08}$, $p = 0.00033$ and $p = 0.00815$. For anti-spike IgA, $p = 0.79$ for all comparisons. Source data are provided as a Source Data file.

**Previous exposure**. The HCW cohort encompassed 23 participants who tested anti-N positive at t0 (excluded from the above analyses). For these participants, anti-RBD Ig levels reached the maximal values of the assay at t1. To increase the dynamic range of the assay, measurements were repeated on diluted samples, revealing a further increase of anti-RBD Ig levels between t1 and t2. Similarly, anti-spike IgG levels reached high values by t1 and increased further at t2 (Fig. 5). IgM responses were not significantly increased, while IgA levels reached values slightly above those of naïve participants at t1 (median [IQR]: 1.51 [1.30 1.85] for N-pos; 1.11 [0.91–1.23] for N-neg), with no enhancement at t2.

**Exposure post 1st vaccine dose**. The HCW cohort encompassed 31 participants who were diagnosed COVID-19 after the 1st vaccination, through epidemiological surveillance by the health system (excluded from the above analyses). For these participants, neither age (median 36 y) or sex (77.4% females) differed when compared to the whole cohort. The day of diagnosis was distributed over the 3 weeks of the interval between the 2 doses, with 24/31 (77.4%) of the infected participants diagnosed during the first 15 days after the 1st dose (Supplementary Fig. S2). COVID-19 containment rules impeded scheduled sampling for antibody measurement, and only 14/31 contributed sera post-infection, at a time point corresponding

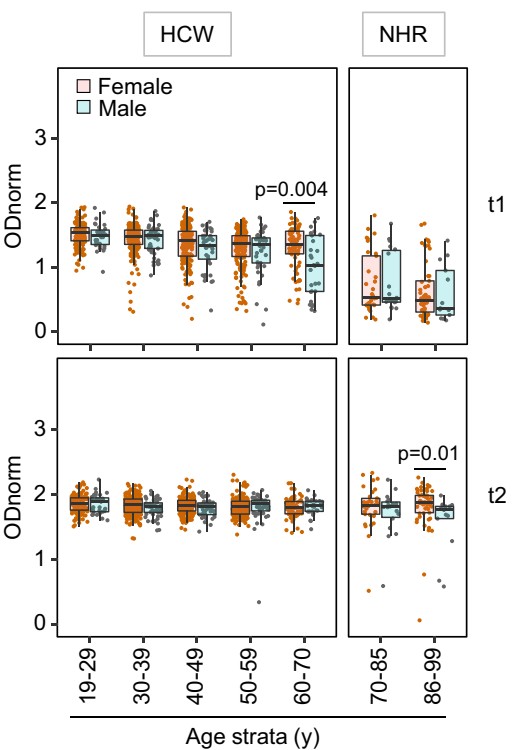

**Fig. 4 Sex modulates vaccine-induced anti-SARS-CoV-2- spike reactivities.** Shown are anti-spike IgG as in Fig. 2b, except that the semi-quantitative measurements are stratified by age and sex in HCW and NHR cohorts at t1 (upper panels) and t2 (lower panels). Sample size: [19–29]y $n = 31M$, 140F; [30–39]y $n = 51M$, 200F; [40–49]y $n = 41M$, 163F; [50–59]y $n = 45M$, 171F; [60–70]y $n = 29M$, 77F; [70–85]y $n = 18M$, 33F; [86–99]y $n = 13M$, 53F. Data points represent individual females (orange) and males (gray), boxes denote interquartile range, horizontal lines represent the median, and whiskers denote the minimum and maximum values below or above the median at 1.5 times the interquartile range. Kruskal–Wallis tests revealed age group differences in the HCW cohort both in males ($p = 0.00122$) and females ($p = 4.96 \times 10^{-09}$) at t1. Wilcoxon rank-sum tests, two-sided, at specific age groups, significant p-values are indicated in the panels, revealed sex differences at t1 in HCW [60–70]y ($p = 0.003749$), and at t2 in NHR [86–99]y ($p = 0.01136$). Source data are provided as a Source Data file.

to t2 (3 weeks post 2nd dose) for the rest of the cohort. Of these, 1 had received a 2nd vaccine dose and showed elevated anti-spike IgG values, 2 could be considered non-responders to the vaccine with specific IgG levels below or at the threshold of positivity, and 11 presented specific IgG levels in the range of that observed at t1 (3 weeks post 1st dose) for the whole cohort, suggesting infection soon after the 1st vaccine dose did not act as a boost.

**Lost to follow-up**. Among the HCW participants who received two vaccine doses and did not develop COVID-19, 220 collected at t0 did not participate in t1 and/or t2 collections (Fig. 1a). We excluded these would be biased to either low or high reactivities as the median and IQR at t0, t1, and t2 for each isotype were in the range of those of the full cohort (supplementary Table 3).

## Discussion

In this study, we found a striking inter-individual variation in the amplitude and nature of the humoral response 3–5 weeks after the 1st vaccine dose, explained only in part by age, sex, previous exposure, and drug treatments. These findings have consequences for our understanding of mRNA vaccine immunogenicity,

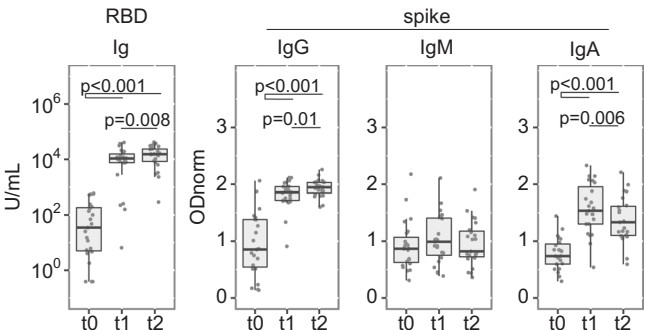

**Fig. 5 Previous exposure to SARS-CoV-2 enhances vaccine-induced anti-SARS-CoV-2- spike reactivities.** Shown are semi-quantitative measurements at t0, t1, and t2 for $n = 23$ participants identified as anti-N positive at t0, prior to BNT162b2 vaccination. Data points represent individual participants, boxes denote interquartile range, horizontal lines represent the median, and whiskers denote the minimum and maximum values below or above the median at 1.5 times the interquartile range. For anti-RBD Ig quantification, samples with a value > or =2500 with the standard ECLIA assay were measured again after a 50x dilution. Quade test for group difference overtime points, p-values $5.383 \times 10^{-10}$ for anti-RBD Ig, $8.013 \times 10^{-10}$ for anti-spike IgG and $1.855 \times 10^{-09}$ for anti-spike IgA, and $p = 0.24$ for anti-spike IgM. Wilcoxon signed-rank test for pairwise comparison, two-sided, with p-value adjustment (Benjamini–Hochberg method), significant p-values indicated in each panel. For anti-RBD Ig, $p = 4.3 \times 10^{-05}$ (t0/t1 and t0/t2) and $p = 0.0081$ (t1/t2); for anti-spike IgG, $p = 5.7 \times 10^{-05}$ (t0/t1 and t0/t2) and $p = 0.014$ (t1/t2); for anti-spike IgA $p = 7.3 \times 10^{-05}$ (t0/t1 and t0/t2) and $p = 0.0059$ (t1/t2). Source data are provided as a Source Data file.

the design of vaccination roll-out and for the management of vaccines.

Our data indicate BNT162b2 elicits a humoral immune response biased to the IgG class with low contribution of IgM and IgA. This result is consistent with classical IgM responses that peak during the first-week post-antigen-encounter, and are not significantly boosted through memory cell recall. Of note, our analysis at t0 reveals a sizable fraction of participants presenting IgM anti-spike reactivity prior to vaccination (12.5% above threshold as compared to 0.8% when testing sera from 1000 donors collected before COVID-19 pandemic), and the nature of these peculiar IgM reactivities remains to be understood. Following infection, strong anti-spike IgA responses are frequent, may be more prevalent in symptomatic patients[19], and confer neutralizing capacity[20]. The rather unchanged anti-spike IgA levels after the 2nd vaccine injection in naïve participants suggest the involvement of T cell independent responses, usually producing monomeric IgA, unlikely to contribute to mucosal immunity upon subsequent SARS-CoV-2 infection. Future work will determine whether, as for IgG[21], specific IgA elicited by mRNA vaccines are present in mucosal secretion.

Identification of a subset of participants testing anti-RBD Ig positive at t0, but unaware of previous exposure to SARS-CoV-2, is consistent with a majority of a/pauci-symptomatic cases remaining undetected during 2020. In this subset, the high IgG responses at t1 confirmed the 1st vaccination dose acts as a boost in individuals previously exposed to SARS-CoV-2[7,8,11,12,22]. This finding strengthens the previous proposition that a single vaccine dose may suffice for maximal antibody response in previously infected individuals[13]. In turn, it is plausible that previously exposed non-symptomatic individuals already had reduced specific-antibody levels at t0 but kept immune memory B cells[18], and these may be the high responders at t1 in the group we classified as naïve.

As observed with conventional vaccines[4], age was a clear factor contributing to the decreased amplitude of the response to the 1st dose. Sex added an effect to age at t1, in the 60–70 strata only, possibly an indirect effect of behavior, co-morbidities or treatments[5]. In contrast, the boosting effect of the 2nd dose was robust in all age strata. A previous work analysing an Italian cohort of health care workers, reported an age effect on IgG antibody titres measured 3 weeks after the 2nd dose of BNT162b2 vaccine[23]. In our study, the slightly lower frequency of sero-converted at t2 in the NHR versus HCW cohort was attributable to the non-responders.

The limitations of the study include the short follow-up upon the 2nd vaccine dose, and the duration of the antibody response will be addressed along the year in subsequent blood sampling for the same cohorts. The study did not include functional assays such as neutralizing antibodies, which have been shown to be predictive of protection from severe disease and to lower extent from infection[24]. However, levels of anti-spike reactivity elicited by BNT162b2 have been previously correlated with in vitro neutralization of spike-pseudoviruses and SARS-CoV-2, including variants of concern, by ourselves and others[9,15,25,26]. Moreover, both binding and neutralizing antibodies have been correlated with mRNA vaccine efficacy[27]. Our study did not include systematic detection of infection cases, and per se does not serve to assess the protection from infection conferred by the 1st or 2nd dose in the time frame of our study. While 31 individuals were diagnosed during the interval between 2 doses, with 77% of these during the first 15 days, unvaccinated controls with similar demographics and occupations at the same time period were not available. A previous work addressing the protection conferred by a single dose of the BNT162b2 vaccine in a cohort of UK health care workers reported no effect up to 13 days, and a 70% reduction in risk of infection 14 days after vaccination[28]. Another work addressing a very large population indicated BNT162b2 vaccine effectiveness was of 56.6% after one dose (14–21 days) and 96.6% after two doses[29], confirming the 2nd dose is required for homogeneity at population level. Ad hoc comparison of vaccine effectiveness in this previous report[29] and seroconversion in our study suggests the threshold of positivity in serological assays may be revised to higher values to infer protection. A more suitable metanalysis will clarify this point.

Those limitations notwithstanding, the low efficacy of single-dose BNT162b2 vaccination makes the compelling argument that a 2nd vaccination dose is required to attain uniformly high levels of immunoglobulin in COVID-19 naive individuals. The large spread in the quality of the antibody response 3–5 weeks after the 1st vaccine dose should be taken into account when considering extending the time between 1st and 2nd administration of the BNT162b2 vaccine. This measure was advocated to optimize vaccine roll-out and population protection in the context of limited COVID-19 vaccine supply (e.g. JCVI-UK prolongation for 12 weeks and NACI Canada up to 16 weeks). Corroborating our concern, 31 participants have diagnosed COVID-19 in between the 2 vaccine doses, possibly the combined result of relaxed precaution measures, the peak of COVID-19 prevalence, and suboptimal immunity. Similar warning emerged from analysis of cancer patients[26]. In support of a scheme of vaccination based on a short prime-boost interval, the boosting responses were similar in participants who received the 2 vaccine doses 3 or 5 weeks apart. Follow-up studies will address the duration of the response elicited by a regimen of 2-dose vaccination with short interval.

Finally, the detection of non-responders by simple reactivity analysis argues for monitoring the post-vaccination antibody level, notably in elderly and immunosuppressed individuals.

Serology tests, possibly point of care, upon vaccination and along time, would guide subsequent measures, such as maintaining social distancing but also considering additional vaccine doses and/or switching to other vaccines containing stronger adjuvant components or a larger number of epitopes.

## Methods

**Recruitment and enrollment**. The two-cohorts study enrolled 1245 healthcare workers (HCW cohort) from three hospitals, administratively grouped in a single regional center (CHLO), in Lisbon, Portugal, and 146 residents at four nursing homes (NHR cohort) in Almeirim, a town located in the vicinity of Lisbon, Portugal. All participants enrolled through volunteer sampling. Participants were scheduled to initiate BNT162b2 mRNA (Pfizer/BioNTech, Comirnaty) vaccination along the original protocol of 2 doses with a 3-week interval. The study was approved by the Ethics committees of the Centro Hospitalar Lisboa Ocidental and the Administração Regional de Lisboa e Vale do Tejo, in compliance with the Declaration of Helsinki, and follows international and national guidelines for health data protection. All participants provided informed consent to take part in the study.

**Blood samples processing and storage**. Venous blood was collected by standard phlebotomy. Blood collection occurred on the day of the 1st vaccination (baseline, t0), the day of the 2nd vaccination (3 weeks later for all HCW and 96/146 NHR, and 4–5 weeks later for 50/146 NHR, t1) and 3 weeks after the 2nd vaccine dose (t2). Serum was prepared using standard methodology.

**Immunoassays**. Electro-chemiluminescence immunoassay (ECLIA) was used to quantify Ig anti-N (Elecsys® Anti-SARS-CoV-2 N, Roche) and anti-RBD (Elecsys® Anti-SARS-CoV-2 S, Roche), ran (on cobas e602) and analysed as per the manufacturer instructions, with a threshold defining positivity at index value = 0.8 U/ml. Where indicated, samples with a value > or =2500 in the standard anti-RBD ECLIA assay were measured again after a 50x dilution. Direct ELISA was used to quantify IgG, IgM and IgA anti-full-length spike. The assay was adapted from Amanat et al.[30]. and semi-automated to measure IgM, IgG and IgA in 384-well format, according to a protocol to be detailed elsewhere. Assay performance was determined by testing 1000 pre-pandemic sera and 40 COVID-19 patients diagnosed at least 10 days prior to sera collection. ROC curve analysis determined a specificity of 99.3%, 99.2%, 99.2%, and a sensitivity of 95.9%, 61.2%, and 73.7% for IgG, IgM, and IgA, respectively. The threshold defining positivity correspond to normalized OD (ODnorm) = 1. Serial titration of 67 COVID-19 patients established the assay has a dynamic range of 3 logs titre.

**Statistics and reproducibility**. The recruitment of healthcare workers and nursing home residents of this observational longitudinal cohort study used a non-probabilistic method, by convenience and volunteer sampling. The experiments were not randomized. Investigators performing the immunoassay and generating ODnorm from raw data were blinded to participant age and sex. No statistical method was used to predetermine sample size. The effect size (Cohen's f, calculated on IgG levels in 5 age groups of the HCW cohort) estimates a minimum group size of $n = 35$ to detect a mean group difference of 0.1 in ODnorm, with a significance level of 0.05 and a power of 80%. No data were excluded from the analyses.

*Missing data management*. Anti-spike antibodies measurements were performed on all participants who adhered to the study (no missing data).

*Reproducibility*. Anti-full-length spike measurements were performed in duplicates, and rare discrepancies between replicates were resolved through repeated duplicate measurements. Data from positive and negative control samples included in each run were reproducible across assays.

Frequency of seroconversion (positivity), anti-spike levels and age effects on these, were successfully replicated with two different immunoassays using two different target antigens: ELISA for anti-full-length-spike and ECLIA for anti-RBD.

Data statistical analyses and graphical design were performed in R (r-project.org), version 4.0.4 GUI 1.74 and R studio, version 1.1.463 and the main package ggplot2 (version 3.3.5) (references in the supplemental material). Continuous variables were summarized using medians and interquartile ranges (IQR). Categorical variables were summarized using frequencies and percentages. For repeated samples, the Quade test was performed to test for differences between strata and the Wilcoxon signed-rank test for pairwise comparison of groups. For independent samples, the Kruskal Wallis test was performed to test for differences between strata, and Wilcoxon rank-sum test for pairwise group comparison with p-value adjustment for multiple testing. All p-values were two-sided, at a significance level of 0.05.

**Reporting summary**. Further information on research design is available in the Nature Research Reporting Summary linked to this article.

## Data availability

Source data are provided with this paper: all data generated in this study (ECLIA and ELISA) and necessary to interpret, verify and extend the research in the article, is provided in the "source data" file. Source data are provided with this paper.

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

## Acknowledgements

We thank the healthcare workers and nursing home residents who participated in the study. We are indebted to Jorge Carneiro for help with ELISA data analysis and Tiago Paixão for guidance in the data processing. We are grateful to Joao Costa and Joana Bom for providing training on highly specialized equipment made available for this work. We acknowledge the serology4covid consortium for joined effort during the implementation of a low-scale pilot version of the ELISA assay. We also thank the healthcare professionals involved in the HCW sample collection and local testing: Inês Sousa, Catarina Farinha, Susana Vaz, Helena Fernandes, Carla Castro, Catarina Simões, Joana Soares, Nara Silva, Ana Matos, Isabel Barros and Inês Santos. We are indebted to the study assistants who ensured the NHR collection: Lara Fontes, Pureza Duarte Ferreira, Paulo Guia, Pedro Silva and João Ferreira. We are most thankful to all the members of the IGC-COVID-19 task force for their continuous support. We thank Antonio Coutinho and Thiago Carvalho for their critical reading of the manuscript. This work benefited from COVID-19 emergency funds 2020 from Calouste Gulbenkian Foundation (to C.P.G. and J.D.) and from Oeiras (to P.M.A.) and Almeirim (to T.N.) city councils. It was also supported by the Science and Technology Foundation, Ministry of Education and Science (FCT, Portugal) through the Project 754-Research4COVID-19– 2nd edition (to C.P.G.) and by PORLisboa 2020, Portugal 2020 and European Union, through European Regional Development Fund through the project FEDER/072558 (to J.D.). The funders had no role in study design, data collection and analysis, decision to publish or preparation of the manuscript.

## Author contributions

J.F.-V., T.N., C.P.G., and J.D. conceived the study. M.-L.B., L.A.G., N.D., C.D., R.C., P.M., L.K., A.O., P.M.A., C.P.-G., and J.D. developed the methods for the study analysis. P.C.B., V.M., and A.B., performed laboratory analysis. J.F.-V., L.A.G., T.P.C., J.M.F., C.C., A.M.C., T.N. and C.P.-G. organized sample acquisition, recruited participants, collected samples, and questionnaires. M-L.B., L.A.G., N.D., C.P.-G., and J.D. carried out data analysis. M.-L.B. performed the statistical analysis. M.-L.B., N.D., C.P.-G., and J.D. co-wrote the manuscript. C.P.G. and J.D. supervised and led the study.

## Competing interests

The authors declare no competing interests.
