## [Peer Review File · Nature Communications]

REVIEWER COMMENTS

Reviewer #1 (Remarks to the Author):

This is a straight forward and well conducted study analyzing the dynamics of antibody development after BNT162b vaccination in health care workers (1254) and nursing home residents (146). The authors main finding is that antibody responses are heterogenous post 1st vaccination dose and normalize on a high level after the 2nd dose accross all age groups. Nevertheless, antibody levels, eg vaccine response, is lower in the elderly. Even though these findings are not entirely new, as similar results have been reported from others with smaller cohorts (the authors quote the studies) the findings are important and highly relevant.

The methodology is sound, the results are clear, the manuscript is well written and the conclusions are supported by the data. On the other hand, there is some overlap with recently published studies, which is not surprising in this highly dynamic field.

<https://pubmed.ncbi.nlm.nih.gov/34140469/>

<https://pubmed.ncbi.nlm.nih.gov/34247945/>

I would like to encourage the authors to mention and discuss those.

Furthermore I think it is important to mention and disucss that RBD-specific antibodies are generally lower in the higher age group; How do the authors explain the finding, that IgA does not show age-specific effects in terms of antibody levels?

Minor:

HCW: health care workers or health care providers, please unify (see Abstract)

Reviewer #2 (Remarks to the Author):

Viana study RBD and Spike antibodies or 2 large cohorts immunised with an mRNA vaccine. They show age dependency and suboptimal responses in NHR. The work is well done and should be of interest to Nature Comms readers.

Comments

The work would be strengthened by analyses of neutralising antibodies, a strong correlate of protection (Khoury, Nature Med 2021).

Fig 2 - RBD Abs should be studied in the NHR.

Fig 2b – the levels of HCW and NHR should be side by side to allow comparison.

The NHR should be stratified by age e.g. < or >85y.o.

Fig 5 – the sample should be diluted further to more accurately quantitate.

Were any of the NHR previously infected?

Further details and antibody analysis of the 32 subjects infected between the doses should be provided. This is of substantial interest and would improve the MS considerably.

The NHR received 2nd dose at 3-5 weeks. Was there an effect on Ab responses (level, or more important, durability) if given at 3 or 5 weeks? This is important.

Point-by-point response to the reviewers' comments

On, Viana et al "Population homogeneity for the antibody response to COVID-19 BNT162b2 / Comirnaty vaccine is only reached after the second dose, across all adult age ranges."

Tracking number: NCOMMS-21-26039A

In this document, replies to reviewers are in **blue fonts**, and the corresponding modified text in the revised manuscript are in *italics*. Underlined are the modifications directly relevant to the specific point addressed. In the revised manuscript file, all changes are highlighted in yellow.

Reviewer #1 (Remarks to the Author):

This is a straight forward and well conducted study analyzing the dynamics of antibody development after BNT162b vaccination in health care workers (1254) and nursing home residents (146). The authors main finding is that antibody responses are heterogenous post 1st vaccination dose and normalize on a high level after the 2nd dose across all age groups. Nevertheless, antibody levels, eg vaccine response, is lower in the elderly. Even though these findings are not entirely new, as similar results have been reported from others with smaller cohorts (the authors quote the studies) the findings are important and highly relevant.

The methodology is sound, the results are clear, the manuscript is well written and the conclusions are supported by the data.

We thank reviewer 1 for their careful revision of the manuscript, positive evaluation and suggestions.

Reviewer 1 point 1 (R1.P1) On the other hand, there is some overlap with recently published studies, which is not surprising in this highly dynamic field.

<https://pubmed.ncbi.nlm.nih.gov/34140469/> <<https://pubmed.ncbi.nlm.nih.gov/34140469/>>

<https://pubmed.ncbi.nlm.nih.gov/34247945/> <<https://pubmed.ncbi.nlm.nih.gov/34247945/>>

I would like to encourage the authors to mention and discuss those.

We concord with the suggestion to update the references lists and appreciate the suggestion for these two works relevant to ours. In the revised version of the manuscript we have included, and commented in the discussion, the two references indicated by R1 as follow:

- **page 9, line 11:**

....In contrast, the boosting effect of the 2nd dose was robust in all age strata. *A previous work analysing an Italian cohort of health care workers, reported an age effect on IgG antibody titres measured 3 weeks after the 2nd dose of BNT162b2 vaccine²³. In our study, the slightly lower frequency of sero-converted at t2 in the NHR versus HCW cohort was attributable to the non-responders.*

Where reference 23 is (<https://pubmed.ncbi.nlm.nih.gov/34247945/>)

- **page 9, line 24 (inserting also a discussion point related to the new result section "EXPOSURE POST 1ST VACCINATION" as per R2 request):**

"....Our study did not include systematic detection of infection cases, and per se does not serve to assess the protection from infection conferred by the 1st or 2nd dose in the time frame of our study. While 31 individuals were diagnosed during the interval between the 2 doses, with 77% of these during the first 15 days, unvaccinated controls with similar demographics and occupations at the same time-period were not available. *A previous work addressing the protection conferred by a single dose of the BNT162b2 vaccine in a cohort of UK health care workers, reported no effect up to 13 days, and a 70% reduction in risk of infection 14 days after vaccination²⁸.*"

Where reference 28 is (<https://pubmed.ncbi.nlm.nih.gov/34140469/>).

R1.P2. Furthermore, I think it is important to mention and discuss that RBD-specific antibodies are generally lower in the higher age group; How do the authors explain the finding, that IgA does not show age-specific effects in terms of antibody levels?

We thank the reviewer for highlighting the findings were not made clear enough. The decrease with age is clearly observed for the anti-RBD bulk response measured in the HCW cohort at t1 (Fig.3 upper left). The anti-RBD assay measures total Ig (as was stated in the manuscript, result section, page 4, line 21: "...Collected samples were analysed for bulk reactivity against SARS-CoV-2-RBD using a

commercial ECLIA ...". A very similar pattern is observed in the same cohort for anti-Spike IgG (Fig.3 upper right), which dominates the antibody response, with IgM and IgA isotypes as minor contributors (Fig.2B and 3). In keeping with anti-spike reactivities, comparison between the HCW and NHR cohorts reveal a clear age effect for all isotypes, including IgA (Fig. 2B). Further age stratification of the NHR cohort (novel panel, see R2P3), revealed IgA decrease with age is only observed between the 60-70 and 70-85 age strata. These additional analyses indicate that the age effect is also true for IgA, albeit less pronounced than for IgG and IgM.

In the revised version, we have rearranged figure 2 to facilitate comparisons between cohorts (complying also with R2.P2) and completed Figure 3 (complying also with R2.P3). We also revise the text in the results section "CUMULATIVE AGE EFFECT" as follow:

- page 6 line 8

"...The parallel analysis of the HCW and NHR cohorts revealed a dramatic effect of advanced age on the magnitude of the antibody response at t1, for all three anti-spike isotypes (Figure 2). Stratified analysis by age groups (in 10-year bins) of the HCW cohort revealed an increasing negative effect of age at t1, evident for bulk anti-RBD reactivity and for anti-spike IgG and IgM, but not IgA levels (Figure 3 and Supplementary Table 2). While age stratification within the NHR cohort ([70-85] and [85-99] years) was not informative, the youngest age strata of the NHR cohort scored lower than the oldest age strata of the HCW cohort, for anti-spike IgG, IgM and IgA (median [IQR] [age strata] for anti-spike IgG: HCW 1.30 [1.00-1.53] [60-69]; NHR 0.50 [0.35-1.06] [70-85]; for anti-spike IgM: HCW 0.71 [0.54-0.97] [60-69]; NHR 0.36 [0.26-0.48] [70-85] and for anti-spike IgA: HCW (1.11 [0.89-1.20] [60-69]; NHR 0.82 [0.59-1.12])."

R1.P3. Minor:

HCW: health care workers or health care providers, please unify (see Abstract)

We thank the reviewer for noticing this discrepancy and we now use "health care workers" throughout the revised text.

Reviewer #2 (Remarks to the Author):

Viana study RBD and Spike antibodies or 2 large cohorts immunised with an mRNA vaccine. They show age dependency and suboptimal responses in NHR. The work is well done and should be of interest to Nature Comms readers.

We thank reviewer 2 for their careful revision of the manuscript and positive evaluation, as well as for the constructive criticisms.

Comments

R2.P1. The work would be strengthened by analyses of neutralising antibodies, a strong correlate of protection (Khoury, Nature Med 2021).

We reckon analysis of neutralising antibodies could be interesting. We did not consider this for several reasons. One them was the size of the cohorts, we analysed more than 3300 samples to reveals the heterogeneity of the Ab response after a 1st dose is normalized following a 2nd vaccine dose. The technical limitation imposed by cellular read-out of neutralization are not compatible with testing the whole cohorts, and neutralization analysis on subsamples would divert from the scope of the paper. Moreover, there is in the literature strong evidences supporting a correlation between neutralization capacity with protection -as noted by R2-, but also between antibody levels and neutralizing capacity, as we had referred to in the discussion. In the revised version, we completed the discussion to encompass the 2 correlations, which together make the argument more compelling, as follow:

- page 9, line 18 :

"...The study did not include functional assays such as neutralizing antibodies, which have been shown to be predictive of protection from severe disease and to lower extent from infection²⁴. However, levels of anti-spike reactivity elicited by BNT162b2 have been previously correlated with *in vitro* neutralization of spike-pseudoviruses and SARS-CoV-2, including variants of concern, by ourselves and others^{9,15,25,26}. Moreover, both binding and neutralizing antibodies have been correlated with mRNA vaccine efficacy²⁷..."

Where reference 24 is (Khoury, Nature Med 2021)

R2.P2. Fig 2 - RBD Abs should be studied in the NHR.

We agree with R2 that analysis of anti-RBD Abs in the NHR cohort would uniformize the displays comparing the two cohorts. We cannot address this request for practical reasons: Anti-RBD antibodies were measured in automated ECLIA equipment (Roche) which requires >200µl of sera/plasma per read. In contrast, our anti-spike Elisa assay uses 4ul for the 3 isotypes, each measured in duplicates. Due to the frailty of this population, only small volume of blood was collected from the NHR participants, not enough to run the anti-RBD Roche test. As mentioned in the reply to reviewer 1 (R1.P2.), the RBD assay measures total Ig and essentially reflects the contribution of the dominant IgG isotype. It seems to us the comparison we provide between the 2 cohorts based on anti-spike IgG, IgM and IgA isotypes provides enough argument to sustain our claims.

R2.P3. Fig 2b ? the levels of HCW and NHR should be side by side to allow comparison.

We appreciate this suggestion that improves the clarity of the figure, and have modified Fig. 2B accordingly.

R2.P4. The NHR should be stratified by age e.g. < or >85y.o.

We thank the reviewer for this suggestion. In the previous version of the manuscript, we had tested the stratification by age of the NHR cohort and solely mentioned but did not shown the results. In the revised version, Figure 3 and 4 now includes age stratification of NHR. The text corresponding to this partitioning was extended in the revised version

page 6 line 12:

“...While age stratification within the NHR cohort ([70-85] and [85-99] years) was not informative, the youngest age strata of the NHR cohort scored lower than the oldest age strata of the HCW cohort, for anti-spike IgG, IgM and IgA...”

R2.P5. Fig 5 ? the sample should be diluted further to more accurately quantitate.

We appreciate the suggestion. We had acknowledged the anti-RBD antibodies at t1 and t2 in the previously infected individuals displayed in Figure 5 were at saturating levels of the assay. For the revised version, we repeated the measurements of anti-RBD antibodies diluting the samples that were saturated. The corresponding panel in Figure 5 was replaced with this data, which now reproduces the increase from t1 to t2 observed for anti-spike IgG. The text in the revised manuscript now states,

- page 7, line 13:

“...For these participants, anti-RBD Ig levels reached the assay maximal values at t1. To increase the dynamic range of the assay, measurements were repeated on diluted samples, revealing a further increase of anti-RBD Ig levels between t1 and t2. Similarly, anti-spike IgG levels reached high values by t1 and increased further at t2.(Figure 5)...”

R2.P6. Were any of the NHR previously infected?

Epidemiologic surveillance by the health system of the NHR cohort prior to vaccination, which is performed periodically in a controlled manner due to the particular susceptibility to severe disease of these individuals, detected no positive cases among residents. Corroborating this survey, none were anti-spike positive at t0. This was described in the result section “ENROLMENT” in the previous version, and now made more explicit in the revised version,

- page 4, line 20:

“In the NHR cohort, epidemiologic surveillance by the health system evidenced no COVID-19 cases prior to vaccination, and all samples collected at day 0 tested negative for anti-spike reactivities.”

R2.P7. Further details and antibody analysis of the 32 subjects infected between the doses should be provided. This is of substantial interest and would improve the MS considerably.

We appreciate the suggestion to present the data we could gather on the participants infected between the doses. Of note, since the first submission of the manuscript additional curation of the database revealed 5 participants (out of 1391) were misclassified along the funneling we present in Fig 1A, and subject infected between the 2 doses are now 31. All data, presented in each figure, table and text, were revised to account for this correction, with no impact on the conclusions.

Analysis of the 31 participants infected between the 2 doses is illustrated in the new supplementary Figure S2, described in the new result section "EXPOSURE POST 1st VACCINE DOSE, and referred to in the discussion as follow,

- page 7, line 20:

"The HCW cohort encompassed 31 participants who were diagnosed COVID-19 after the 1st vaccination, through epidemiological surveillance by the health system (excluded from above analyses). For these participants, neither age (median 36 y) or sex (77,4% females) differed when compared to the whole cohort. The day of diagnosis was distributed over the 3 weeks of the interval between the 2 doses, with 24/31 (77.4%) of the infected participants diagnosed during the first 15 days after the 1st dose (Supplementary Figure S2). COVID-19 containment rules impeded scheduled sampling for antibody measurement, and only 14/31 contributed sera post-infection, at a time point corresponding to t2 (3 weeks post 2nd dose) for the rest of the cohort. Of these, 1 had received a 2nd vaccine dose and showed elevated anti-spike IgG values, 2 could be considered non-responders to the vaccine with specific IgG levels below or at the threshold of positivity, and 11 presented specific IgG levels in the range of that observed at t1 (3 weeks post 1st dose) for the whole cohort, suggesting infection soon after the 1st vaccine dose did not act as a boost."

- page 9, line 23:

"...Our study did not include systematic detection of infection cases, and per se does not serve to assess the protection from infection conferred by the 1st or 2nd dose in the time frame of our study. While 31 individuals were diagnosed during the interval between 2 doses, with 77% of these during the first 15 days, unvaccinated controls with similar demographics and occupations at the same time-period were not available. A previous work addressing the protection conferred by a single dose of the BNT162b2 vaccine in a cohort of UK health care workers, reported no effect up to 13 days, and a 70% reduction in risk of infection 14 days after vaccination²⁸. Another work addressing a very large population indicated BNT162b2 vaccine effectiveness was of 56.6% after one dose (14-21 days) and 96.6% after two doses²⁹, confirming the 2nd dose is required for homogeneity at population level..."

R2.P8 The NHR received 2nd dose at 3-5 weeks. Was there an effect on Ab responses (level, or more important, durability) if given at 3 or 5 weeks? This is important.

We appreciate the suggestion to present data addressing the effect of the duration of the interval between 1st and 2nd dose on the Ab responses. These are now presented in Supplementary Figure S1, in a new result section "EXTENDED PRIME-BOOST INTERVAL", and referred to in the discussion. The impact on the durability of the antibody response can only be assessed through analysis at later timepoints, which is out of the scope of this manuscript, though a mention is made to this point in the discussion. The text in the revised version now reads

- Page 7 line 2:

"The NHR participants received the 2nd vaccine dose either 3 (n=44) or 5 (n=74) weeks after administration of the 1st dose. In both sub-groups, the 2nd dose boosted the anti-spike IgG levels to high values, with an apparent higher amplitude after the shorter time interval, and resulted in similar frequency of positivity (94,1% for 3-, and 93,2% for 5-week interval) (Supplementary Figure S1). As a possible confounding factor, sex distribution was different between the two groups, with more males in the 5-weeks group (82% females in the 3- and 69% in the 5-weeks group), while the average age was similar. This data supports the notion that a shorter prime-boost interval does not prevent a robust recall response."

- Page 10, line 15:

"...In support to a scheme of vaccination based on a short prime-boost interval, the boosting responses were similar in participants who received the 2 vaccine doses 3 or 5 weeks apart. Follow-up studies will address the duration of the response elicited by a regimen of 2-dose vaccination with short interval..."

REVIEWERS' COMMENTS

Reviewer #1 (Remarks to the Author):

all concerns/comments were adequately addressed

Reviewer #2 (Remarks to the Author):

I am satisfied with the response.

Response to reviewers about the revised version

Manuscript NCOMMS-21-26039A

Population homogeneity for the antibody response to COVID-19 BNT162b2 / Comirnaty vaccine is only reached after the second dose across all adult age ranges.

Reviewer #1 (Remarks to the Author):

all concerns/comments were adequately addressed

We thank Reviewer #1 for their constructive criticisms at the 1st reviewing step, which contributed to improve the manuscript.

Reviewer #2 (Remarks to the Author):

I am satisfied with the response.

We thank Reviewer #2 for their constructive criticisms 1st reviewing step, which contributed to improve the manuscript.

Sincerely yours,

Jocelyne Demengeot
Corresponding author